# Exogenous Cannabinoids Impair Effort-Related Decision-Making via Affecting Neural Synchronization between the Anterior Cingulate Cortex and Nucleus Accumbens

**DOI:** 10.3390/brainsci13030413

**Published:** 2023-02-27

**Authors:** Zahra Fatahi, Mohammad Fatahi, Mirmohammadali Mirramezani Alizamini, Ahmad Ghorbani, Mohammad Ismail Zibaii, Abbas Haghparast

**Affiliations:** 1Neuroscience Research Center, Shahid Beheshti University of Medical Sciences, School of Medicine, Tehran P.O. Box 19615-1178, Iran; 2School of Electrical and Computer Engineering, College of Engineering, University of Tehran, Tehran P.O. Box 14395-515, Iran; 3CAS Key Laboratory of Mental Health, Institute of Psychology, Beijing 100101, China; 4Department of Psychology, University of Chinese Academy of Sciences, Beijing 100049, China; 5Laser and Plasma Research Institute, Shahid Beheshti University, Tehran P.O. Box 19839-6941, Iran; 6School of Cognitive Sciences, Institute for Research in Fundamental Sciences, Tehran P.O. Box 19395-5531, Iran

**Keywords:** cannabinoid, effort-based decision-making, neural synchronization, anterior cingulate cortex, nucleus accumbens, field potential recording

## Abstract

Humans and animals frequently make an endeavor-based choice based on assessing reinforcement value and response costs. The cortical-limbic-striatal pathway mediates endeavor-based choice behavior, including the nucleus accumbens (NAc) and the anterior cingulate cortex (ACC). Furthermore, cannabinoid agonists demonstratively impairs decision-making processes. In this study, neural synchronization and functional connectivity between the NAc and ACC while endeavor-related decision-making and reaching reward were evaluated. The effect of exogenous cannabinoids on this synchronization was then assessed. A T-maze decision-making task with a differential expense (low vs. high endeavor) and remuneration (low vs. high remuneration) was performed and local field potentials (LFP) from the ACC and NAc were registered simultaneously. Results showed functional connectivity during endeavor-related decision-making while the animals chose the high endeavor/high remuneration in both regions’ delta/beta (1–4 and 13–30 Hertz) frequency bands. Furthermore, functional connectivity existed between both areas in delta/theta (1–4 and 4–12) frequencies while reaching a remuneration. However, neural simultaneity was not observed while the animals received cannabinoid agonists, making a decision and reaching remuneration. The obtained results demonstrated that functional connectivity and neural simultaneity between the NAc and ACC in delta/beta and delta/theta frequencies have a role in endeavor-related decision-making and reaching remuneration, respectively. The effect of exogenous cannabinoids on decision-making impairment is relevant to changes in the ACC and NAC brain wave frequencies.

## 1. Introduction

To make a successful decision, comparing the costs of a given option to its related benefits is required [1]. As an essential element for modulating synaptic action, endocannabinoids regulate an extensive range of brain actions such as memory, pain processing, and stress reaction. They have also emerged as a crucial regulator of effort-based decision-making [2,3,4]. The endocannabinoid system is a neural substrate implicated in decision-making behavior and partly affects decision-making by inhibiting the activity of inhibitory and excitatory neurotransmission along the mesocorticolimbic pathway [5]. Indicatively, both systemic injection and administering into the NAc of a cannabinoid receptor agonist disrupted effort-related decision-making behavior [6,7,8].

Acute cannabis use is related to the impairment of attention, memory, learning, the ability to plan, solve problems, and make decisions, and this causes users to make risky decisions that they would not otherwise make [9]. A recent study about young adults revealed that cannabis use was related to an enhanced risk of suicidal ideation, suicidal plan, and suicide attempts [10]. Studies also showed that chronic cannabis use is associated with an enhanced risk of psychiatric illness, cognitive difficulties, and addiction [11]. Cannabinoid receptors are richly expressed in the nucleus accumbens (NAc) and anterior cingulate cortex (ACC) shown as two essential brain areas in effort-based decision-making [12,13,14,15]. The NAc has a vital role in motivation and reward, and dysfunction in this area is one of the reasons for anhedonia and social withdrawal [16]. The ACC has an essential role in information processing while making a decision. Furthermore, a considerable decrease was observed in a functional pathway projecting from the ACC to the NAc in cannabis users compared to non-using controls [17,18]. A wide range of studies demonstrated strong connections between the ACC and NAc [19,20]. Significant projections from the ACC to the core of the NAc and inactivation and/or lesion of the ACC results in rats choosing low endeavor/low remuneration options [13,20,21,22].

Evidence shows the rhythmic coupling in various brain tasks such as attention, memory, and decision-making [23,24,25]. A previous study revealed that cannabinoid agonists in the NAc disrupted delay- and effort-related decision-making [6,7]. Administration of the cannabinoid agonist into the ACC changed cost-benefit decision-making such that rats prefer low remuneration/low endeavor instead of high remuneration/high endeavor [25]. On the other hand, a neural and functional connection between the OFC and ACC was established during endeavor- and delay-related decision-making [26,27]. Therefore, knowing the efficacy of exogenous cannabinoids in the ACC and NAc on value-related decision-making and considering serial functions and projection between the ACC and NAc, this study aimed to explore the efficacy of exogenous cannabinoids in the NAc on the functional connection of the ACC and NAc during endeavor-related decision-making. To fulfill this aim, before and administrating cannabinoids within the NAc core simultaneously, this study investigated local field potential (LFP) from the ACC and NAc core when rats performed a T-maze cost-benefit specimen.

## 2. Materials and Methods

### 2.1. Animal

Male Wistar rats (230–270 g; Pasture Institute, Tehran, Iran) were maintained under lights on at 07:00, and lights off at 19:00; temperature 22 ± 2 °C. Food was adjusted for initial body weights of about 85–90% of the free-feeding weight while the beginning of the experiment and after this, a controlled weight gain of about 6–12 g per week and water was available ad libitum. Guide for the care and use of Laboratory Animals (National Institutes of Health Publication No. 80-23, revised 1996) was used employed for all the experiments. Furthermore, approval was given by the Institutional Ethics Committee of the Shahid Beheshti University of Medical Sciences (IR.SBMU.PHNS.REC.1398.136), Tehran, Iran, to the experiments.

### 2.2. Drugs

In the current study, these drugs were used: AM251, as a cannabinoid receptor antagonist, was attenuated in 12% DMSO (Tocris Bioscience, Bristol, UK). Win 55,212-2, as a cannabinoid receptor agonist (Tocris Bioscience, Bristol, UK), was solved in 12% dimethyl sulfoxide (DMSO; Sigma Aldrich, Taufkirchen, Germany; was diluted in saline).

### 2.3. Apparatus

A T-maze endeavor-related decision-making task was used [27]. The maze was made of Plexiglas and had three arms (two target arms and one start arm; each arm was ten cm wide, 60 cm long, and 40 cm high). Food wells were placed at the end of the target arms (Appendix A).

### 2.4. Behavioral Training

Rats were trained to T-maze decision-making tasks with a differential (low endeavor vs. high endeavor) and remuneration (low vs. high remuneration) in the two target arms [27,28]. Before starting training, the animals were set on a restricted nutrition program. When they achieved 85–90% of their free-nutrition weight, the animals entered the T-maze.

Habituation phase: In the beginning, the rats were located at the start arm of the T-maze (three rats together), and for 20 min, they were free to explore all over the maze. In the next three days, single rats investigated the maze while plentiful food was in both nutrition wells of the target arms (45 mg food remuneration, Formula A/I; P. J. Noyes, Lancaster, NH).

Discrimination phase: As the habituation phase finishes, it is time to learn how to discriminate between low-rewarded (two food remunerations) and high-rewarded (six food remunerations) target arms. Infrared sensors were located at the end of the start arm, the beginning of the target arms, and the beginning of food wells. Once the animals break the infrared sensor at the end of the start arm, an audio signal is heard from a speaker at the end of the high remuneration arm to determine the high remuneration arm for the animals. The high remuneration arm was randomized; in half of the trials high remuneration arm was on the right and in the other half, the trials were on the left. Three steps of discrimination training are as below:

1. As the high remuneration arm contains six pieces of remuneration and the low remuneration one has two pieces, the rats were placed at the beginning arm, and without any barrier between the target arms, they were free to get the remuneration from both arms. This step lasted for three days; every trial (by each rat) was ten times a day.

2. One of the target arms was closed in each trial, and the rats achieved one of those target arms. They did ten daily trials to complete the second step, which lasted three days. To test the target arms, in five trials, rats accessed only to low remuneration arm, and for the rest half, they accessed the high remuneration arm.

3. In the third step, the rats were permitted to gather food only from one of the target arms, using ten “choice” and two “force” trials every day with two minutes gap between trials (all were concluded consecutively). The training continued until the animals selected the highly rewarded arm in 80% (for three consecutive days). After consuming the remuneration, the condition for removing the rats from the maze was choosing one of the target arms.

Barrier phase: To boost the physical endeavor cost of the high remuneration, as soon as the rats received the average rate of 80% for high remuneration, a ten cm high barrier (made of wire mesh) was put in the middle of the high remuneration arm. Likewise, in this phase, when the animals chose over 80% (average high remuneration choice, HRC) for three consecutive days, the barrier’s height reached 30 cm (with 10 cm steps) at the end of the training process (Appendix A).

### 2.5. Experimental Design

After surgery recovery, rats performed two forced and ten selection trials of the endeavor-related decision-making T-maze task per day till the animals gained the pre-surgery execution rate for three following days. On test day, after the forced decision trials, rats (*n* = 8) received 12% DMSO into the NAc core bilaterally, and after five minutes, they were placed on the T-maze and accomplished ten selection trials (all experiments were performed between 10 a.m. until 15:00 p.m.). The percentage of choice of high remuneration/high endeavor and/or low remuneration/low endeavor arm was recorded for each rat. The second group of rats obtained bilateral (0.3 μL/side) intra-NAc core microinjection of efficient dose of Win 55,212-2 (50 and μM/0.3 μL DMSO) [6]. Five minutes after Win 55,212-2 microinjection, each rat completed ten selection trials, with a distance of ~2 min, and the percentage of high remuneration/high endeavor and/or low remuneration/low endeavor choice was assessed. The third group of rats received an effective dose of cannabinoid antagonist, AM251 (45 μM/0.3 μL DMSO) [6], before microinjection of Win 55,212-2 (50 μM) and performed ten choice trials (Appendix A).

There are five infrared sensor beams (IR) (one was at the end of the start arm, two were at the beginning of high remuneration and low remuneration arms, and two others were at the beginning of food wells; Appendix A). Local field potential recording (LFP) was recorded simultaneously from the ACC and NAc core during each trial in all groups. Once the rat reaches the start arm’s end and breaks the infrared sensor beam, an audio signal is heard to determine the high remuneration arm. When the animals entered the target arm, they broke the infrared sensor at the beginning of the target arm. Five hundred msec before and five hundred msec after breaking an infrared sensor beam at the beginning of the target arm were analyzed while investigating neural function during decision-making time. In addition, when the animals broke the infrared sensor beam at the beginning of the food wells (Appendix A), five hundred msec before and five hundred msec after that were analyzed to investigate neural activity while reaching a remuneration. Trials were video recorded timestamp the animal’s movement with electrophysiology recordings.

Another control experiment was performed to assess the spontaneous movement activity of the animals following the administration of cannabinoids into the NAc. Three separate groups of rats were investigated (*n* = 8); the first group received DMSO 12%, the second group received an agonist, and the third group received the antagonist + agonist intra NAc core. Five minutes after administration, the animals were located in the middle of an open field measuring 30 cm × 30 cm with 30 cm walls and were permitted to travel freely inside the field for 20 min. A 3CCD camera registered the locomotor activity (Panasonic Inc., Kadoma, Japan) mounted 2 m above the open field and was analyzed offline using Ethovision video tracking software (version 3.1, Noldus Information Technology, Wageningen, The Netherlands).

To investigate the conceivable involvement of spatial preference or memory in the rats’ decisions, they were trained on a control task in which an equivalent 30 cm height obstacle was newly located also to the low remuneration arm (equal effort). The training followed for about ten days to ensure that the changed rule was well stable for rats. On the test days, rats obtained administration of an efficient dose of Win 55,212-2 or DMSO (0.3 μL/per side) in the NAc core in an equilibrium manner, and behavioral factors were investigated as above.

### 2.6. General and Surgical Procedures

The intraperitoneal administration of ketamine (100 mg/kg) and xylazine (8 mg/kg) for anesthetizing was conducted to implant the electrodes and cannula. Animals were then immobilized in a stereotaxic device (Stoelting Co., Wood Dale, IL, USA). Two winded PFA-coated stainless-steel wires constructed bipolar recording electrode wires (diameter: 0.005 in altogether ~200 μm distance between the tips). The electrodes were placed separately into the left ACC and NAC core, and the cannula was placed into the NAc core bilaterally. The rat brain atlas was used for the determination of coordinates [29]. For the ACC, 1.2 mm anterior to bregma, 0.7 mm lateral to the midline, and 2.6 mm ventral to the skull, and for the NAC core: 0.9 mm anterior to bregma, ±1.7 mm lateral to the midline and 6.9 mm ventral to the skull surface (Appendix A).

A 5-pin mini Molex plug (13.5 × 3 × 7 mm, 0.2 g) was connected to wires. Then, using acrylic cement (Vertex, Boston, MA, USA), they were mounted to the skull (*n* = 8 rat). Dental acrylic cement (Vertex, MA, USA) was used to firm the grounding wire and electrodes.

### 2.7. Neural Recording and Wavelet Analysis

To demonstrate a time-series signal in a time-frequency framework the Continuous wavelet transform (CWT) is used. A translated version of the original wavelet function ψ_0_ and the scaled time-series signal can be represented as a convolution product. The original wavelet function has different scales that are shrunken and stretched in time. It is required that the original wavelet be translated several times along the time axis to represent all lengths of the signal because its shrunken version cannot represent the whole length of the time-series signal. The transform of the continuous wavelet with a time series x(n), (*n* = 1, 2, 3, …, *N*), which is sampled from a continuous sample at a time step Δ*t* is described as:$$W^{X}\left(n.s\right)=\sqrt{\frac{\Delta t}{s}}\overset{N}{\underset{n^{\prime }=1}{\sum }}x_{n^{\prime }}\psi_{0}\left[\left(n^{\prime }-n\right)\frac{\Delta t}{s}\right]$$

Here, *n* shows the translation factor, and *s* is the scale factor which is inversely proportional to the frequency. The similarity between the translated/scaled versions of the original wavelet and signal is shown by the CMT coefficient.

The coherence of the wavelet calculates the degree of correlation between two time-series signals in distinct frequencies, and time ranges can be defined according to the CWT. The coherence of the squared cross-wavelet is written as:$$R^{2}\left(n.s\right)=\frac{\left|S\left[S^{-1}W^{XY}\left(n.s\right)\right]\right|}{S\left[S^{-1}{\left|W^{X}\left(n.s\right)\right|}^{2}\right].S\left[S^{-1}{\left|W^{Y}\left(n.s\right)\right|}^{2}\right]}$$

The cross wavelet-transform for two time-series *y(n)* and *x(n)* is defined by $W^{XY}=W^{X}W^{Y*}$, and the complex conjugate is shown by ***.

Here, *S* is used to show a good measure of coherence as a smoothing operator.

The coherency between two signals (time series) is shown by $R^{2}\left(n.s\right)$ and it is limited between 0 and 1. The correlation of two time-series signals in a distinct time and frequency scale can be the interpretation of the squared cross-wavelet. The analytic morlet wavelet is considered a proper way for the computation of the time-frequency coherence demonstration because brings a proper balance between frequency and time localization [30].

The coherence of the wavelet Is counted outside the cone of influence (COI) and because the regions inside the COI are influenced by the edge effects then regions inside COI probably are not valid. To get a significance level for the coherence, $R^{2}\left(n.s\right)$, the magnitude of the surrogate data was compared with of coherence of the real data [31]. Bootstrapping was used for creating a big (*n* = 1000) number of surrogate data pairs and then for each of those data pairs the coherence was calculated. *p* < 0.05 was considered for the significance of the compared simulated and real data pairs. A commercial acquisition processor (Niktek, http://niktek.ir/ (accessed on 11 July 2014)) was used to record, filter, and digitalize the neural activities for extracting LFPs (at the sampling rate of 1000 Hertz). Analysis of wavelet coherence was carried out by wavelet toolbox of Matlab software version 2018a (The MathWorks, Inc., Natick, MA, USA)

#### Wavelet Analysis

A wavelet is a wave-like oscillation that has a finite duration and its average is zero:(1)∫ψ(t)dt=0x

Based on ψ(t), which is called the “mother wavelet”, other functions can be generated:(2)ψa,b(t)=1aψ(t−ba), a,b∈ℝ

The constants a and b are called scale and translation, respectively. The functions generated by Equation (2) are the same functions as Equation (1) which are translated and dilated/shrunk.

The continuous wavelet transform (CWT) of a time-series function x[n] is defined as:$$W^{X}(n,s)=\sqrt{\frac{\Delta t}{s}}\sum_{n=1}^{N}x_{n^{\prime }}\psi \left[\left(n^{\prime }-n\right)\frac{\Delta t}{s}\right]x$$

For a constant translation n and scale S, the CWT coefficient represents the similarity between the function x[n] and ψ(n,s). The higher the similarity, the greater the CWT coefficients, and vice versa. Using CWT, it is possible to measure the degree of coherence in different scales and time points between two functions. The squared cross-wavelet coherence is defined as:(3)$$R^{2}(n,s)=\frac{\left|S\left[S^{\left(-1\right)}W^{XY}(n,s)\right]\right|}{\left(S[S^{\left(-1\right)}{\left|W^{X}(n,s)\right|}^{2}].S\left[S^{\left(-1\right)}{\left|W^{Y}(n,s)\right|}^{2}\right]\right)}$$

Equation (3) S is a smoothing operator used to give a useful measure of coherence. $R^{2}(n,s)$ measures the coherence between two time-series signals. It can be interpreted as the correlation between two signals in specific time and scale ranges. The magnitude of squared cross-wavelet coherence ranges between 0 and 1; 0 means a lack of coherence and 1 means complete coherence.

Recordings, digitalization, and filtering of neural activities were conducted using a mercantile acquisition processor (Niktek, http://niktek.ir/ (accessed on 11 July 2014)). Recordings were bandpass-filtered (0.01–250 Hertz) to extract LFPs and then sampled at 1000 Hertz. Wavelet coherence analysis was carried out using the wavelet toolbox of Matlab software 2018a (The MathWorks, Inc., Natick, MA, USA). Time-Frequency correlation representations were computed using the analytic morlet wavelet which is a good selection since it provides a good balance between time and frequency localization [30]. The magnitude of the correlation of real data was compared with the magnitude correlation of surrogate data to obtain the significance level of correlation, $R^{2}(n,s)$. In particular, a large number (*n* = 1000) of surrogate data pairs were manufactured using bootstrapping and then the correlation was calculated for each of these data pairs. The correlation of the real data pair was compared with all of these simulated data pairs and was considered significant in regions where *p* < 0.05.

Of course, it should be noted that there are a number of caveats that are commonly encountered in LFP applications. There are four common issues that warrant caution with respect to the exegesis of connection assessments that we were unable to address in this study: (1) the signal-to-noise ratio problem: in LFP recordings the signals that are used for the connection assessment consist of a mixture of a main signal and noise. (2) The common reference problem, (3) the volume conduction/field spread problem, and (4) unobserved common input. In LFP recordings, counterfeit functional connection assessments can result from the application of a common reference channel. Visualize two recorded time series, data 1 and data 2. Each of these signals consists of the difference in the electric potential measured at the position of the electrode and the position of the reference electrode. If the same reference electrode is applied for both electrodes that are subsequently applied for the connection assessment, the fluctuations in the electric potential at the reference position will be reflected in both time series, yielding fictitious connectivities at a zero time lag. Any connectivity metric that is sensitive to correlations at a zero time lag will in part be spurious [32].

### 2.8. Statistics

One-way analysis of variance (ANOVA) was followed by a posthoc Newman–Keuls test, and a t-test was applied as a requirement. P-values of less than 0.05 (*p* < 0.05) were considered to be statistically significant. The data were analyzed by GraphPad Prism® 5.0.

## 3. Results

### 3.1. Effects of the Administration of Exogenous Cannabinoid in the NAc Core on High Remuneration Choice (HRC) Percentage in Endeavor-Related Decision-Making the Task

One-way ANOVA followed by post hoc Newman–Keuls test [F(2,25) = 97.31, *p* < 0.0001] clarified that administrating cannabinoid agonists (Win 55,212-2, 50 μM ) caused a substantial attenuation in HRC percentage. Microinjection of AM251 (CB1R antagonist, 45 μM) before administration of Win 55,212-2 prevented the efficacy of CB1R agonist on HRC (Figure 1). Furthermore, the results of the measurement of locomotor activity [F(2,23) = 0.2275, *p* = 0.7985] demonstrated that in agonist- and antagonist-treated animals, distanced travel did not have any significant efficacy compared to the DMSO-received group. Therefore, the efficacy of cannabinoid on decision-making following intra-NAc administration are not due to hypoactivity or hyperactivity (Appendix A).

In addition, this study trained the Win 55,212-2-received rats on an additional equivalent control task to assess the conceivable involvement of spatial preference or memory in the decisions of rats. In this task, there was an equal expense (30 cm barrier), but different remuneration (6 pellets vs. 2 pellets) in both the hinge and low remuneration arm. Results [t = 0.7206, df = 12, *p* = 0.4850] clarified that in the equivalent control task, Win 55,212-2-treated animals preferred high/endeavor remuneration, and no significant decrease was found in high remuneration choice percent compared to the DMSO-control group in this task (Appendix A).

### 3.2. A High Degree of Correlation between the ACC and NAc Core in the DMSO-Control Group While Making an Endeavor-Related Decision and Choosing a High Endeavor/High Remuneration Arm

In DMSO-received animals, rats chose the high remuneration/high endeavor arm in almost every trial. Once the animals broke IR at the beginning of the target arm, 500 ms before and 500 ms after that, LFP signals were analyzed to investigate neural synchronization between ACC-NAc core during endeavor-related decision-making. Figure 2A shows the coherence between the ACC and NAc of the group average of high remuneration/high endeavor trials in the control group while making an endeavor-based decision and choosing a high remuneration/high endeavor arm. An enhancement oscillation has occurred after poking the IR gate in 2–4 frequencies. Figure 2D shows the areas where the coherence is significant compared to surrogate data (*n* = 1000, *p* < 0.05) using bootstrapping. Based on the figure, an alteration is observed in significant coherence after poking the IR gate. The change in frequency range can also be verified by looking at the group average z-normalized time-domain signals of ACC and NAC areas. As seen in Figure 3A, the ACC signal’s amplitude decreases when the stimulus begins, while the stimulus does not significantly affect the NAC signal (Figure 3D). Lower oscillations are related to low-frequency components, so the time domain can justify signals’ coherence in the low-frequency range after poking the IR gate.

### 3.3. A Poor Correlation between the ACC and NAc Core in the Cannabinoid Agonist-Treated Group While Making an Endeavor-Related Decision and Choosing Low Endeavor/Low Remuneration Arm

In agonist-received animals, rats chose a low remuneration/low endeavor arm in almost 80% of trials. Based on Figure 2B, no coherence appears between ACC and NAc areas in a low-frequency range in agonist-treated animals while choosing the low remuneration/low endeavor arm. The areas in which coherence is significant are shown in Figure 2E, where bootstrapping (*n* = 1000, *p* < 0.05) has been used to distinguish significance. Figure 3B,E show the z-normalized time-domain signal of the ACC and NAC areas. No discernible change existed after beginning the stimulus.

### 3.4. A High Correlation between the ACC and NAc Core in the Antagonist-Treated Group While Making an Endeavor-Based Decision and Choosing a High Remuneration/High Endeavor Arm

In antagonist-received animals, rats chose the high remuneration/high endeavor arm in almost 85% of trials. Analysis of these trials indicated a powerful correlation between ACC and NAc areas in the low-frequency range (2–4, as shown in Figure 2C,F) while making an endeavor-based decision. The results of bootstrapping (*n* = 1000) and then comparing the simulated data with accurate data reveal a significant power (*p* < 0.05). Figure 3C,F show the z-normalized time-domain signal of ACC and NAc, respectively. There is a discernible change in the dynamics of the signals.

Furthermore, a comparison of correlation values in control, agonist-treated, and antagonist-treated groups was performed during the 1000 ms time window. A repeated measures one-way ANOVA, followed by posthoc Kruskal-Wallis tests [43.53, *p* < 0.0001] clarified that in control and antagonist-received groups, functional activity in delta and beta was remarkably higher than in the agonist-received group (Figure 4).

In addition, it was performed directly compared coherence between the groups (control vs. CB1 agonist vs. CB1 agonist + antagonist). For decision-making time, repeated measures one-way ANOVA followed by post hoc Kruskal-Wallis tests [13.49, *p* = 0.0012] showed that in control and antagonist-received groups, functional activity was higher than in the agonist-received group.

### 3.5. A High Degree of Correlation between the ACC and NAc Core in the DMSO-Control Group While Reaching the Remuneration

Once the animals broke IR at the beginning of food wells, 500 ms before and 500 ms after that, LFP signals were analyzed to investigate neural synchronization between ACC-NAc core while reaching a remuneration. Figure 5A shows the coherence between the ACC and NAc of the group average of high remuneration/high endeavor trials in the control group while reaching the remuneration. An enhancement oscillation has occurred before and after poking the IR gate in delta and theta frequencies. Figure 5D shows the areas where the coherence is significantly compared to surrogate data (*n* = 1000, *p* < 0.05) using bootstrapping. Based on the figure, significant coherence areas were altered before and after poking the IR gate. The change in frequency range can also be verified by looking at the group average z-normalized time-domain signals of ACC and NAC areas. As seen in Figure 6D, the NAc signal’s amplitude decreases as soon as the stimulus begins, while the stimulus does not significantly affect the ACC signal (Figure 6A). Lower oscillations are related to low-frequency components, so the time domain can justify signals’ coherence in the low-frequency range after poking the IR gate.

### 3.6. Poor Correlation between the ACC and NAc Core in the Agonist-Treated Animals While Reaching the Remuneration

Figure 5B, shows no coherence between ACC and NAC areas in a low-frequency range in the agonist-treated group while reaching the remuneration. The areas in which coherence is significantly shown in Figure 5E that again, bootstrapping (*n* = 1000, *p* < 0.05) has been used to distinguish significance. Figure 6B,E show the z-normalized time-domain signal of the ACC and NAC areas. There is not any discernible change after beginning the stimulus.

### 3.7. A High Correlation between the ACC and NAc Core in the Antagonist-Treated Group While Reaching the Remuneration

Analysis of antagonist-treated trials indicates a powerful coherence between ACC and NAC areas in delta and theta (Figure 5C,F) while reaching the remuneration. The results of bootstrapping (*n* = 1000) and then comparing the simulated data with actual data reveal a significant power (*p* <0.05). Figure 6C,F show the z-normalized time-domain signal of ACC and NAC, respectively. There is a discernible change in the dynamics of the signals.

Furthermore, the correlation values in control, agonist-received, and antagonist-received groups were compared during the 1000 ms time window. A repeated measures one-way ANOVA, followed by post hoc Kruskal-Wallis tests [KW = 34.63, *p* < 0.0001], clarified that functional activity in delta and theta frequency bands in control and antagonist-treated groups was remarkably higher than in the agonist-treated group (Figure 7).

In addition, it was performed directly compared coherence between the groups (control vs. CB1 agonist vs. CB1 agonist + antagonist). For reaching to reward time, repeated measures one-way ANOVA, followed by post hoc Kruskal-Wallis tests [KW = 7.799, *p* = 0.0202], clarified that functional activity in the control and antagonist-treated groups was higher than in the agonist-treated group.

## 4. Discussion

Here, it has been intended to investigate the efficacy of exogenous cannabinoids on endeavor-related decision-making, which is conducted through regulating ACC-NAc neural synchronization. To this purpose, an effective dose of cannabinoid receptor agonist and antagonist was administered prior to the T-maze task. The LFP of both NAc core and ACC was recorded during the behavior session. Results illustrated *(I)* an enhanced oscillation between the NAc core and ACC while making an endeavor-related decision in DMSO-control animals. *(II)* The experiments also showed that coherence between the NAc core and ACC was changed during endeavor-related decisions in agonist-treated animals. *(III)* Furthermore, a high degree of correlation was demonstrated between the ACC and NAc core while reaching the remuneration in control animals, which was altered in the agonist-treated group.

Control rats preferred the high remuneration/high endeavor arm such that the percentage of high remuneration selection was about 90% in these animals. LFP results revealed that when rats prefer the high remuneration/high endeavor arm, frequency bands between the NAc core and ACC in delta and beta frequencies (1–4 and 13–30 Hertz) are enhanced. In fact, in the control group, in high remuneration/high endeavor trials, there is neural synchronization during decision-making from 300 ms before until 100 ms after breaking the IR gate at the beginning of the goal arm. However, in low remuneration/low endeavor, this synchronization was not observed (Appendix A). In the agonist-treated group, rats preferred the low remuneration/low endeavor arm in almost 80% of trials; they were willing to choose low remuneration without physical endeavor instead of high remuneration with the physical endeavor. Accordingly, the increase of delta/beta frequencies did not observe in these animals. However, the administration of cannabinoid antagonists before agonist injection increased delta/beta frequency bands, such as the control group. Furthermore, rats chose the high remuneration/high endeavor arm in almost 85% of trials. On the other hand, obtained data demonstrated a neural synchronization between the ACC and NAc core in delta and theta (1–4 and 4–12) frequencies while reaching a remuneration. Microinjection of cannabinoid agonist into the NAc core destroyed this synchronization; however, antagonist injection returned it, such as the control group. Indeed, in both control and antagonist groups, when the animals chose the high endeavor/high remuneration arm, the neural connectivity between the NAc core and ACC was increased while making an endeavor-based decision choice and reaching a remuneration.

The policy alteration of a number of countries to legitimatize the use of cannabis has resulted in a reduction of the percept of the hazard of using cannabis and an enhancement in its use from teenagers to pregnant women [33]. Recent data revealed that cannabis use more than doubled from 2017 to 2019. A number of documents showed that individuals who have used cannabis regularly before age 16 had dearths on standard neurocognitive tests and one study of cannabis users age 35 or older demonstrated significantly poorer cognitive domains of attention and working memory [9]. Previous studies have mentioned the involvement of both NAc and ACC in endeavor-related decision-making [34]. Dopamine receptors in the ACC regulate endeavor-related choice [15], and ACC-lesioned animals could not maximize session-bundle utility after price/budget changes, showing the disruption of higher-order choice-strategy adaptations [35] Stimulation of dopamine receptors in the NAc attenuated the choice of high endeavor option and enhanced latency of choice significantly [36]. Another study revealed that the NAc core, but not the shell, is part of a neural circuit that mediates endeavor-related selection behavior [37]. The previous findings suggested that activating the cannabinoid system in the NAc disturbed endeavor-related decision-making and caused the animals were less intending to invest in the physical endeavor to gain large remuneration [6]. The NAc showed remarkably increased connectivity with the ACC during high-risk decision behaviors [38]. A significant positive functional connectivity between the NAc and ACC and recorded LFP revealed an enhanced phase-phase coupling of alpha oscillations between the NAc and ACC [39,40]. Furthermore, this study established that the ACC-NAc core coherence was decreased in cannabinoid-treated animals compared to DMSO-control and antagonist-treated animals. Therefore, the disruption of value-based decision-making in cannabinoid-treated animals may be partly related to the alteration of coherence and functional connectivity between the ACC-NAc core. In other words, in the control group, during endeavor-related choice, neural simultaneity between the ACC and NAc core designates the result of the decision partly, leading to preferring high remuneration/high endeavor. Exogenous cannabinoid affects this neural synchronization, causing to prefer low remuneration without physical endeavor.

The NAc is one of the brain centers mediating reinforcement processes and behavioral effects of remuneration [41,42]. Evidence clarified the entity of cannabinoid receptors in the NAc. It has been revealed that the NAc is involved in drug-associated remuneration and has an essential role in the memory of the original remuneration stimulus [43,44]. This study showed a neural coherence between the ACC and NAc in delta and theta frequency bands in the DMSO-control and the antagonist-treated groups while reaching remuneration. However, the administration of cannabinoid agonists into the NAc core impaired this coherence. Therefore, neural synchronization between the ACC-NAc is different when the animals reach a high remuneration with physical endeavor compared to when they reach a low remuneration without physical endeavor. When rats chose to reach the high endeavor/high remuneration arm, delta, and theta frequencies increased. However, this increase was not observed when selecting to reach the low remuneration/low endeavor arm in agonist-treated animals. It is worth mentioning there are common issues that warrant caution with respect to the exegesis of connection assessments (as mentioned in neural recording and wavelet analysis) that we were unable to address in this study [32].

Taken together, the current study demonstrated that the neural simultaneity and coherence between the NAc core and ACC in delta/theta and theta/beta frequency bands have a critical role in the effort-choice decision and reaching a remuneration, respectively. Additionally, impairment in value-related decision-making in cannabis uses is partly associated with the change in neural coherence and synchronization between these regions.

## Figures and Tables

**Figure 1 brainsci-13-00413-f001:**
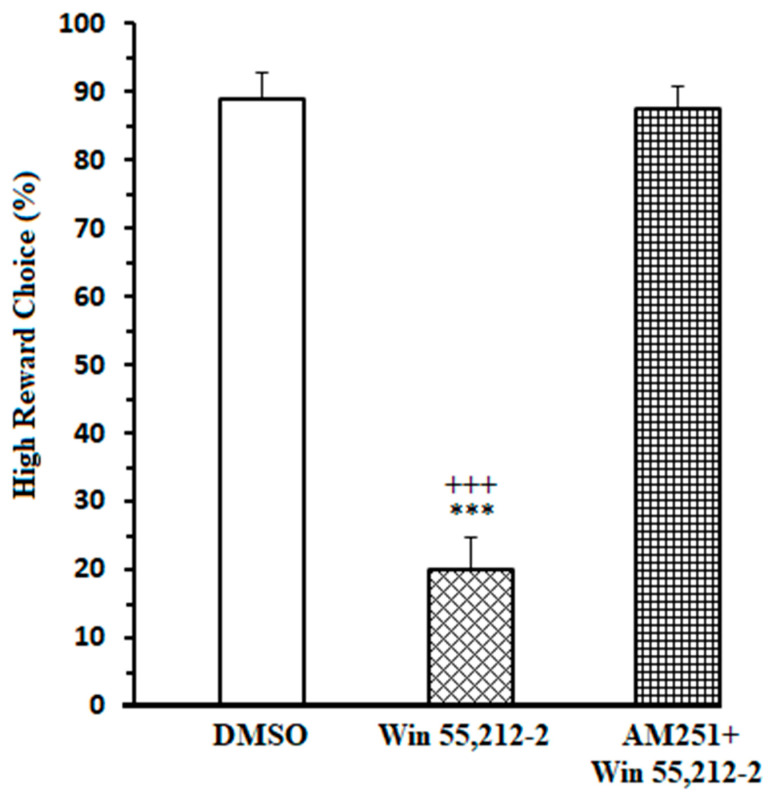
Microinjection of an effective dose of CB1R agonist (Win55, 212-2; 2, 50 µM) into the NAc core decreased the high remuneration choice percentage. Administration of an effective dose of CB1R antagonist (AM251, 45 µM) before microinjection of agonist (50 µM) prevented the agonist efficacy. Data are shown as mean ± SEM for eight rats. *** *p* < 0.001 different from the control group. +++ *p* < 0.001 different from the CB1 receptor antagonist + agonist group.

**Figure 2 brainsci-13-00413-f002:**
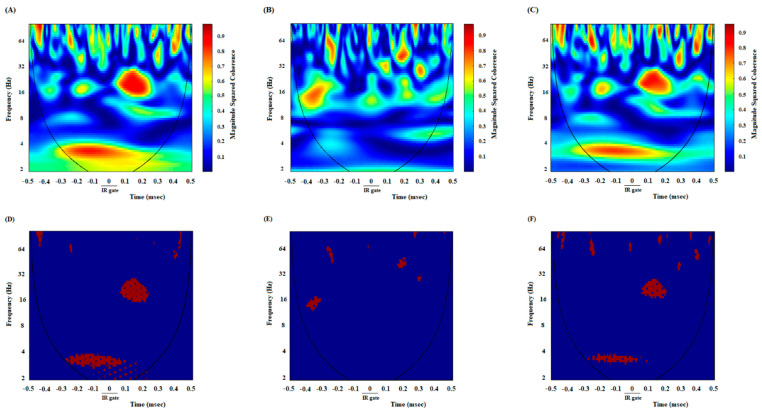
Comparison of coherence of group averages between high remuneration/high endeavor trials in the control group (**A**), low remuneration/low endeavor trials in the agonist-treated group (**B**), and high remuneration/high endeavor trials in antagonist-treated animals (**C**) during endeavor-related decision-making. Time-frequency representation of group averages coherence between the ACC and NAc for *n* = 53 trials. The cone of influence (COI) where edge effects should be considered is blurred. Notably, the wavelet scales are converted to approximate frequencies on the graphs. Comparison of areas in which the coherence is significantly different compared to a large number of surrogate data between high remuneration/high endeavor trials in the control group (**D**), low remuneration/low endeavor trials in agonist-treated animals (**E**)**,** and high remuneration/high endeavor trials in antagonist-treated animals (**F**). The red regions in the figures show the areas with a significant difference (*p* < 0.05). Surrogate data were simulated using bootstrapping (*n* = 1000).

**Figure 3 brainsci-13-00413-f003:**
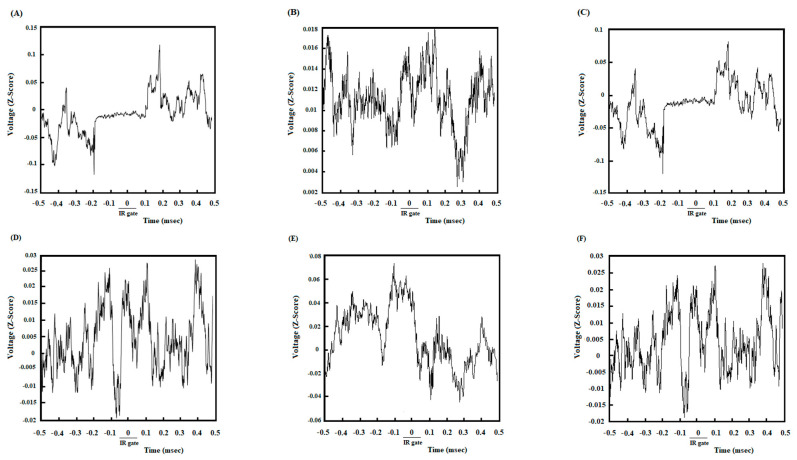
Z-normalized time-domain signals of ACC. The group’s time-domain averages signal acquired from ACC in the control group (**A**), agonist-treated group (**B**), and antagonist-treated group (**C**). In control and antagonist-received groups, a large difference in the time domain dynamics as soon as cutting the infrared sensor happens and lasts for about 0.3 s. In particular, the amplitude of oscillations is decreased significantly. Z-normalized time-domain of group averages signals acquired from the NAc. The group’s time-domain averages signal acquired from NAc in the control group (**D**), agonist-treated group (**E**), and antagonist-treated group (**F**).

**Figure 4 brainsci-13-00413-f004:**
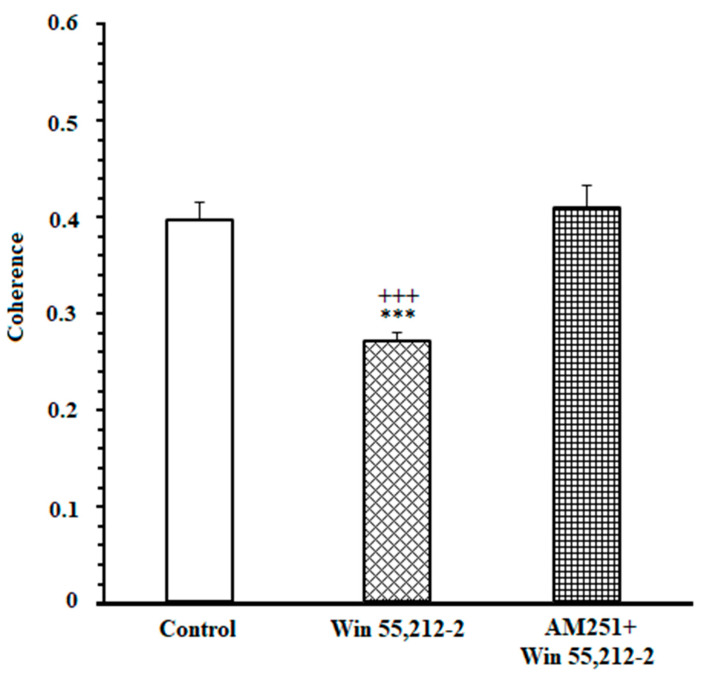
Mean ± SD of coherence in 1000 ms time window (500 ms before and 500 ms after cutting the beam of the infrared sensor). *** *p* < 0.001 different from the control group. +++ *p* < 0.001 different from the CB1 receptor antagonist + agonist group.

**Figure 5 brainsci-13-00413-f005:**
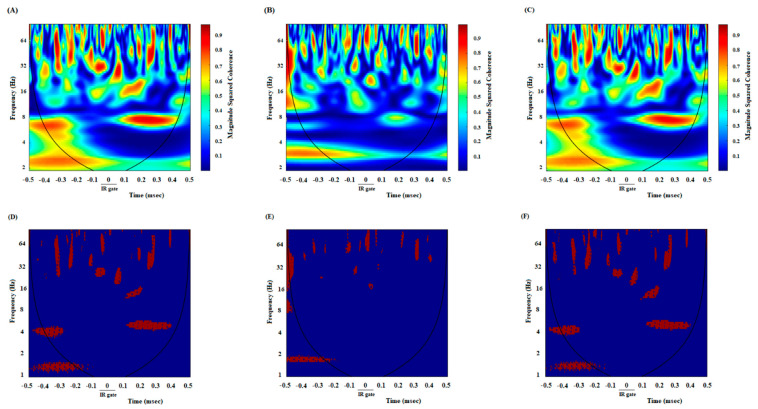
Comparison of coherence of group averages between high remuneration/high endeavor trials in the control group (**A**), low remuneration/low endeavor trials in the agonist-treated group (**B**), and high remuneration/high endeavor trials in the antagonist-treated group (**C**) during reaching remuneration. Time-frequency representation of group averages coherence between the ACC and NAc for *n* = 53 trials. The cone of influence (COI) where edge effects should be considered is blurred. It should be noted that the wavelet scales are converted to approximate frequencies on the graphs. Comparison of areas in which the coherence is significantly different compared to a large number of surrogate data between high remuneration/high endeavor trials in the control group (**D**), low remuneration/low endeavor trials in the agonist-received-group (**E**), and high remuneration/high endeavor trials in the antagonist-received group (**F**) while reaching a remuneration. The red regions in the figures show the areas with a significant difference (*p* < 0.05). Surrogate data were simulated using bootstrapping (*n* = 1000).

**Figure 6 brainsci-13-00413-f006:**
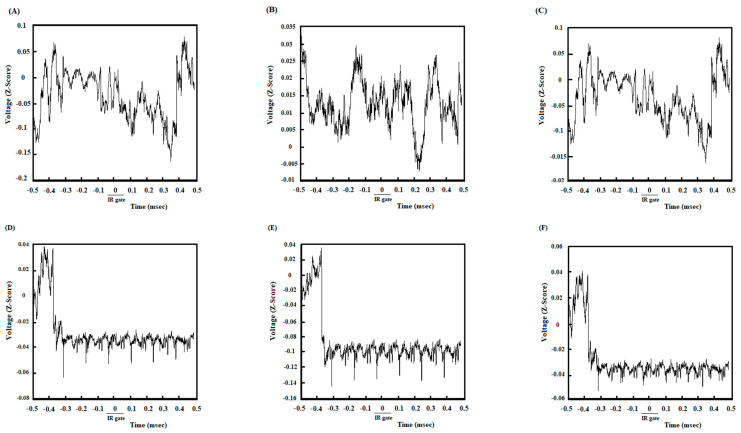
Z-normalized time-domain signals of ACC. The group’s time-domain averages signal acquired from ACC in control animals (**A**), agonist-treated animals (**B**), and antagonist-treated animals (**C**) while reaching a remuneration. Z-normalized time-domain of group averages signals acquired from the NAc. The group’s time-domain averages signal acquired from NAc in the control group (**A**), the agonist-treated animals (**B**), and the antagonist-treated animals (**C**) while reaching a remuneration. The group’s time-domain averages signal acquired from NAc in the control group (**D**), the agonist-treated group (**E**)**,** and the antagonist-treated group (**F**).

**Figure 7 brainsci-13-00413-f007:**
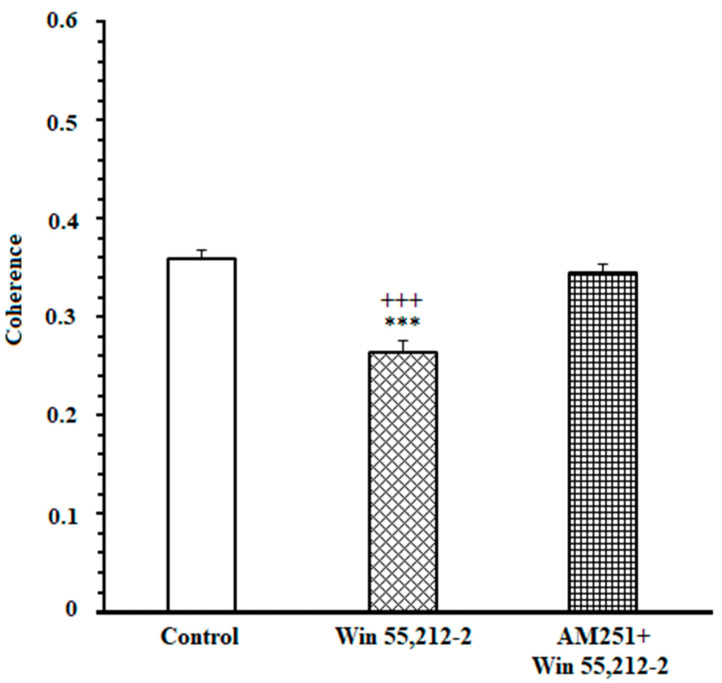
Mean ± SD of coherence in 1000 ms time window (500 ms before and 500 ms after cutting the infrared sensor). *** *p* < 0.001 different from the control group. +++ *p* < 0.001 different from the CB1 receptor antagonist + agonist group.

## Data Availability

Data will be made available upon request. The datasets generated during and/or analyzed during the current study are not publicly available, but are available from the corresponding author on reasonable request.

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
