# Peer review of "Exogenous Cannabinoids Impair Effort-Related Decision-Making via Affecting Neural Synchronization between the Anterior Cingulate Cortex and Nucleus Accumbens"

_brainsci, 2023, doi:10.3390/brainsci13030413_

Round 1
Reviewer 1 Report
In this manuscript, the authors demonstrate that agonism of CB1 receptors in the NAc core biases choice towards low effort, low value options in a decision-making task. Furthermore, the authors demonstrate an increase in functional connectivity between the anterior cingulate and nucleus accumbens core both at the time of decision and at the time of reward receipt during the decision-making task. They then demonstrate that the CB1 agonist disrupts this functional connectivity. The findings are very interesting and have the potential to shed light on the neurobiology underlying the influence of the cannabinoid system on effortful decision-making. However, there are a number of concerns that need to be addressed.
In supplemental Figure 2, all of the "Correct" placements in the nucleus accumbens core are actually in the anterior commissure, while 4 of the 5 "Misplacements" are actually in the nucleus accumbens core. Can the authors demonstrate that there are no significant differences in their LFP analyses when comparing anterior commissure placements to accumbens core placements? Otherwise, the authors will have to restructure their conclusions taking the anterior commissure into consideration. Even if there are no differences, the authors should clearly state that some of the placements were in the anterior commissure and they should include the "Misplacements" that were actually in the accumbens core in their analyses.
The authors show that the CB1 agonist causes animals to choose the small low-effort reward more. Furthermore, the CB1 agonist reduces ACC-NAc synchrony at the choice point. However, the link between ACC-NAc synchrony and choice is not clear. Was there a correlation between ACC-NAc synchrony and choice of the high vs. low reward? Such a correlation would do more to suggest a link between ACC-NAc synchrony and effortful choice.
The authors' analysis of LFPs is rooted in two papers specializing in geophysics (references 27 and 28). While there is likely a great deal of overlap in these methods and those commonly employed in neuroscience, there are a number of caveats and pitfalls that are commonly encountered in LFP applications (e.g. volume conduction) that do not appear to be taken into consideration in this paper. The authors should address or at the very least acknowledge these limitations in their own work; a good reference for these is the following:
Bastos AM, Schoffelen JM. A tutorial review of functional connectivity analysis methods and their interpretational pitfalls. Frontiers in systems neuroscience. 2016 Jan 8;9:175.
The authors assess significance by comparing coherence in the real data to bootstrapped surrogate data. While this certainly will demonstrate where coherence increases in comparison to baseline, it does not directly compare coherence between the groups (control vs. CB1 agonist vs. CB1 agonist+antagonist). The authors need to make a statistically direct comparison before they can make any conclusions about the change in coherence between groups.
There are a number of issues with Figures 3 and 6 that should be addressed. First, the y-axes should be the same across all graphs, otherwise it is difficult to compare across graphs. Second, since these are averages of z-normalized time-domain signals, there should also be error bars indicating standard error of the mean to allow us to see the magnitude of the effect relative to the variance. Third, in both figures, Panels A and C appear to be identical, and Panels D and F appear to be identical. Did the authors accidentally used the same data for these panels? I would expect them to be similar, but not identical.
Supplemental Figure 1B is confusing and should be changed. It starts off as the experimental timeline, describing training and surgery procedures for the animals. However, partway through it switches into becoming a task timeline, describing different events in the task. These two timelines should be separated and described independently.
In Figures 2 and 5 the authors make reference to "correct" and "error" trials when they presumably mean "high effort" and "low effort" trials. They should either make this change or explain what they mean by "correct" and "error".
At the end of the first paragraph of the introduction (lines 46-47), the authors state "It was shown that the microinjection of a cannabinoid receptor agonist into rats disrupted effort-related decision-making behavior [6-8]". If they're using the word "microinjection" they should note the region it's being injected into. In the case of references 6-7, this is the nucleus accumbens, but reference 8 is a systemic injection.
Throughout the manuscript there are several grammar errors that should be fixed.
Author Response
Dear Prof. Stephen D. Meriney,
Thank you and the respectful reviewers for making such constructive comments about the manuscript. We have modified the manuscript accordingly and have added all the necessary explanations to the article based on the points raised by the reviewers, which are all now marked in blue color. We would appreciate any other modifications and suggestions raised by referee and editorial board.
Reply to Reviewer 1# comments:
In supplemental Figure 2, all of the "Correct" placements in the nucleus accumbens core are actually in the anterior commissure, while 4 of the 5 "Misplacements" are actually in the nucleus accumbens core. Can the authors demonstrate that there are no significant differences in their LFP analyses when comparing anterior commissure placements to accumbens core placements? Otherwise, the authors will have to restructure their conclusions taking the anterior commissure into consideration. Even if there are no differences, the authors should clearly state that some of the placements were in the anterior commissure and they should include the "Misplacements" that were actually in the accumbens core in their analyses.
- Thank you very much for your constructive comment. There has been some careless in determining the correct locations. Placement in anterior commissure are misplacement, and these data were out of analysis. Now, we determined correct and mistake places correctly in the “supplementary figure 2”. Thank you again.
The authors show that the CB1 agonist causes animals to choose the small low-effort reward more. Furthermore, the CB1 agonist reduces ACC-NAc synchrony at the choice point. However, the link between ACC-NAc synchrony and choice is not clear. Was there a correlation between ACC-NAc synchrony and choice of the high vs. low reward? Such a correlation would do more to suggest a link between ACC-NAc synchrony and effortful choice.
- Thank you very much for your pertinent comment. We analyzed low reward/low effort trials in the control group, and showed it in “supplementary figure 4”. In high reward/high effort trials, there is neural synchronization during decision-making from 300 ms before until 100 ms after breaking IR gate at the beginning of goal arm. However, in low reward/low effort trials this synchronization was not observed. We added this point in “ Discussion” section in the main text.
The authors' analysis of LFPs is rooted in two papers specializing in geophysics (references 27 and 28). While there is likely a great deal of overlap in these methods and those commonly employed in neuroscience, there are a number of caveats and pitfalls that are commonly encountered in LFP applications (e.g. volume conduction) that do not appear to be taken into consideration in this paper. The authors should address or at the very least acknowledge these limitations in their own work; a good reference for these is the following:
Bastos AM, Schoffelen JM. A tutorial review of functional connectivity analysis methods and their interpretational pitfalls. Frontiers in systems neuroscience. 2016 Jan 8;9:175.
- Thank you very much for your comment. Some explanation about caveats and pitfalls that are commonly encountered in LFP applications was added in “6. Neural recording and wavelet analysis” section in the main text.
The authors assess significance by comparing coherence in the real data to bootstrapped surrogate data. While this certainly will demonstrate where coherence increases in comparison to baseline, it does not directly compare coherence between the groups (control vs. CB1 agonist vs. CB1 agonist+antagonist). The authors need to make a statistically direct comparison before they can make any conclusions about the change in coherence between groups.
- According your comment, we directly compared coherence between the groups (control vs. CB1 agonist vs. CB1 agonist+antagonist). For decision-making time, repeated measures one-way ANOVA followed by post hoc Kruskal-Wallis tests [13.49, P=0.0012] showed that in control and antagonist-received groups, functional activity was higher than in the agonist-received group. For reching to reward time, repeated measures one-way ANOVA followed by post hoc Kruskal-Wallis tests [KW=7.799, P=0.0202] clarified that functional activity in control and antagonist-treated groups was higher than in the agonist-treated group.
There are a number of issues with Figures 3 and 6 that should be addressed. First, the y-axes should be the same across all graphs, otherwise it is difficult to compare across graphs. Second, since these are averages of z-normalized time-domain signals, there should also be error bars indicating standard error of the mean to allow us to see the magnitude of the effect relative to the variance. Third, in both figures, Panels A and C appear to be identical, and Panels D and F appear to be identical. Did the authors accidentally used the same data for these panels? I would expect them to be similar, but not identical.
- Thank you very much for your pertinent comment. According to your comment, the y-axes became the same across all graphs. About error bar, showing error bar of averages of z-normalized time-domain signals in matlab cannot be done, areas in which the coherence is significantly different were shown in Fig, 2D,E,F and Fig. 5D,E,F. In addition, we checked all data again. Data of each panel are different, data of panel A and D were for DMSO administration, and data of panel C and F were for antagonist administration.
Supplemental Figure 1B is confusing and should be changed. It starts off as the experimental timeline, describing training and surgery procedures for the animals. However, partway through it switches into becoming a task timeline, describing different events in the task. These two timelines should be separated and described independently.
- Thank you for your comment. We provided two separated timelines and described them independently in the supplementary figure 1B.
In Figures 2 and 5 the authors make reference to "correct" and "error" trials when they presumably mean "high effort" and "low effort" trials. They should either make this change or explain what they mean by "correct" and "error".
- Thank you very much for your pertinent comment. Correct trial means high effort/high reward trial, and error trial means low effort/low reward trial. We improved it in figures 2 and 5, and mentioned high remuneration/high endeavor trials instead of correct trials, and low remuneration/low endeavor trials instead of error trials.
At the end of the first paragraph of the introduction (lines 46-47), the authors state "It was shown that the microinjection of a cannabinoid receptor agonist into rats disrupted effort-related decision-making behavior [6-8]". If they're using the word "microinjection" they should note the region it's being injected into. In the case of references 6-7, this is the nucleus accumbens, but reference 8 is a systemic injection.
- Thank you for your comment. We modified it in the main text, and mentioned that “both systemic injection and administering into the NAc of a cannabinoid receptor agonist disrupted effort-related decision-making behavior”.
Throughout the manuscript there are several grammar errors that should be fixed.
- Thank you for your comment. The manuscript was edited for proper English language, grammar, punctuation, spelling, and overall style by one of the highly qualified subject-expert native English speaking editors. In addition, there are some other change in the main text which is just to decrease of similarity.
We would appreciate any other modifications and suggestions raised by referee and editorial board.

Reviewer 2 Report
Thank you for this study. Relevant clinical issues may be the only question worth addressing, even if hypothetical. For example, the use of cannabis in the age group it is so commonly used--school-age children. Questions about marijuana and amotivational syndrome seem less relevant with these data than priorities and shaped reinforced behaviors. Cannabis, as a medicine, is widely discussed and the APA Council has reviewed ( Hill et al, AJP 12/8/21 ) but was missing a discussion of the issues raised by your paper.
Author Response
Dear Prof. Stephen D. Meriney,
Thank you and the respectful reviewers for making such constructive comments about the manuscript. We have modified the manuscript accordingly and have added all the necessary explanations to the article based on the points raised by the reviewers, which are all now marked in blue color. We would appreciate any other modifications and suggestions raised by referee and editorial board.
Reply to Reviewer 2# comments:
Thank you for this study. Relevant clinical issues may be the only question worth addressing, even if hypothetical. For example, the use of cannabis in the age group it is so commonly used--school-age children. Questions about marijuana and amotivational syndrome seem less relevant with these data than priorities and shaped reinforced behaviors. Cannabis, as a medicine, is widely discussed and the APA Council has reviewed (Hill et al, AJP 12/8/21) but was missing a discussion of the issues raised by your paper.
- Thank you very much for your pertinent comment. According to your comment, we added some explanations about clinical effects of cannabis and also cannabis use in adolescence period in “ Introduction and 4. Discussion” sections in the main text.
We would appreciate any other modifications and suggestions raised by referee and editorial board.
Round 2
Reviewer 1 Report
The authors responded to some, but not all of my comments. Furthermore, the response to one of my comments about electrode placements was confusing and I would like clarification since it could have a large impact on the results of the paper. The authors’ responses (in italic text) and my new comments are below:
Thank you very much for your constructive comment. There has been some careless in determining the correct locations. Placement in anterior commissure are misplacement, and these data were out of analysis. Now, we determined correct and mistake places correctly in the “supplementary figure 2”. Thank you again.
The correct and misplaced electrode tips are now correctly identified in Supplementary Figure 2. However, I do not understand why there was no change to the results. If some of the prior “correct” electrodes were actually misplacements, and some of the prior “misplacement” electrodes were actually correct, there should now be different electrodes used for the analysis. Because different data is used, the results should not be identical to the previous results using the wrong electrodes. However, many of the figures and results are identical to the previous results that used the incorrect electrodes. Can the authors explain how this is possible? Are these results using the old electrodes with incorrectly reported placements?
Thank you very much for your comment. Some explanation about caveats and pitfalls that are commonly encountered in LFP applications was added in “6. Neural recording and wavelet analysis” section in the main text.
Did the authors make any efforts to address these caveats? If not, the authors should state at line 263: “There are four common issues that warrant caution with respect to the exegesis of connection assessments that we were unable to address in this study:” The authors should also note this as a limitation in their interpretation of the LFP findings in the discussion section.
According your comment, we directly compared coherence between the groups (control vs. CB1 agonist vs. CB1 agonist+antagonist). For decision-making time, repeated measures one-way ANOVA followed by post hoc Kruskal-Wallis tests [13.49, P=0.0012] showed that in control and antagonist-received groups, functional activity was higher than in the agonist-received group. For reching to reward time, repeated measures one-way ANOVA followed by post hoc Kruskal-Wallis tests [KW=7.799, P=0.0202] clarified that functional activity in control and antagonist-treated groups was higher than in the agonist-treated group.
I could not find these statistics in the revised manuscript. Could the authors put them in the manuscript?
Thank you very much for your pertinent comment. According to your comment, the y-axes became the same across all graphs. About error bar, showing error bar of averages of z-normalized time-domain signals in matlab cannot be done, areas in which the coherence is significantly different were shown in Fig, 2D,E,F and Fig. 5D,E,F. In addition, we checked all data again. Data of each panel are different, data of panel A and D were for DMSO administration, and data of panel C and F were for antagonist administration.
The y-axes are still different across all the graphs in Fig. 3 and Fig. 6. Please give them the same range (for example, -0.2 to 0.1 volts) or explain why this cannot be done.
Thank you for your comment. We provided two separated timelines and described them independently in the supplementary figure 1B.
These timelines are current both labeled Figure 1B. One of them should be labeled Figure 1C.
Finally, the authors removed Supplementary Figure 3 from the folder with the Supplementary figures. Was this accidental? They still reference it in the text. They do not reference Supplementary Figure 4, which they should do.
Author Response
Dear Prof. Stephen D. Meriney,
Thank you and the respectful reviewer for making such constructive comments about the manuscript. We have modified the manuscript accordingly and have added all the necessary explanations to the article based on the points raised by the reviewers, which are all now marked in blue color. We would appreciate any other modifications and suggestions raised by referee and editorial board.
Reply to Reviewer 1# comments:
The correct and misplaced electrode tips are now correctly identified in Supplementary Figure 2. However, I do not understand why there was no change to the results. If some of the prior “correct” electrodes were actually misplacements, and some of the prior “misplacement” electrodes were actually correct, there should now be different electrodes used for the analysis. Because different data is used, the results should not be identical to the previous results using the wrong electrodes. However, many of the figures and results are identical to the previous results that used the incorrect electrodes. Can the authors explain how this is possible? Are these results using the old electrodes with incorrectly reported placements?
- Thank you for your comment. During data analysis, data with tip of electrodes in the NAc core was analyzed as main data, and data with tip of electrode in the other sites (such as the NAc shell and anterior commissure) considered as misplacements and was out of analysis. Therefore, all data analysis was from data with tip of electrode in the NAC core. However, there was careless in showing correct places and misplacements in the supplementary figure 2, which according to your previous comment, it was improved.
Thank you very much for your comment. Some explanation about caveats and pitfalls that are commonly encountered in LFP applications was added in “6. Neural recording and wavelet analysis” section in the main text. Did the authors make any efforts to address these caveats? If not, the authors should state at line 263: “There are four common issues that warrant caution with respect to the exegesis of connection assessments that we were unable to address in this study:” The authors should also note this as a limitation in their interpretation of the LFP findings in the discussion section.
- Thank you for your comment. We mentioned “There are four common issues that warrant caution with respect to the exegesis of connection assessments that we were unable to address in this study” in “6. Neural recording and wavelet analysis” section in the main text. Besides, we noted this point in “4. Discussion” section in the main text.
According your comment, we directly compared coherence between the groups (control vs. CB1 agonist vs. CB1 agonist+antagonist). For decision-making time, repeated measures one-way ANOVA followed by post hoc Kruskal-Wallis tests [13.49, P=0.0012] showed that in control and antagonist-received groups, functional activity was higher than in the agonist-received group. For reching to reward time, repeated measures one-way ANOVA followed by post hoc Kruskal-Wallis tests [KW=7.799, P=0.0202] clarified that functional activity in control and antagonist-treated groups was higher than in the agonist-treated group. I could not find these statistics in the revised manuscript. Could the authors put them in the manuscript?
- Thank you for your pertinent comment. We put in in “ Results” section in the main text.
Thank you very much for your pertinent comment. According to your comment, the y-axes became the same across all graphs. About error bar, showing error bar of averages of z-normalized time-domain signals in matlab cannot be done, areas in which the coherence is significantly different were shown in Fig, 2D,E,F and Fig. 5D,E,F. In addition, we checked all data again. Data of each panel are different, data of panel A and D were for DMSO administration, and data of panel C and F were for antagonist administration. The y-axes are still different across all the graphs in Fig. 3 and Fig. 6. Please give them the same range (for example, -0.2 to 0.1 volts) or explain why this cannot be done.
- Thank you for your comment. We modified figure 3 and 6, and gave y-axes same range between -0.2 to 0.2. New figures 3 and 6 exist in folder.
Thank you for your comment. We provided two separated timelines and described them independently in the supplementary figure 1B. These timelines are current both labeled Figure 1B. One of them should be labeled Figure 1C.
- Thank you for your pertinent comment. We modified it in supplementary figure 1.
Finally, the authors removed Supplementary Figure 3 from the folder with the Supplementary figures. Was this accidental? They still reference it in the text. They do not reference Supplementary Figure 4, which they should do.
- Thank you for your comment. Supplementary figure 3 was not removed from the manuscript, it has not any changed after revise the manuscript. Therefore, we did not add it in the folder of response to reviewers. Supplementary figures 1 and 2 have changed, and supplementary figure 4 is new, so we added just them in the folder of response to reviewers.
We would appreciate any other modifications and suggestions raised by referee and editorial board.
